# Using a UAV Thermal Infrared Camera for Monitoring Floating Marine Plastic Litter

**Lonneke Goddijn-Murphy** [1,*], **Benjamin J. Williamson** [1], **Jason McIlvenny** [1] **and Paolo Corradi** [2]

1. Environmental Research Institute, North Highland College, University of the Highlands and Islands, Thurso KW14 7EE, UK; benjamin.williamson@uhi.ac.uk (B.J.W.); jason.mcilvenny@uhi.ac.uk (J.M.)
2. European Space Research and Technology Centre, European Space Agency, 2200 AG Noordwijk, The Netherlands; paolo.corradi@esa.int
* Correspondence: lonneke.goddijn-murphy@uhi.ac.uk

**Abstract:** In recent years, the remote sensing of marine plastic litter has been rapidly evolving and the technology is most advanced in the visible (VIS), near-infrared (NIR), and short-wave infrared (SWIR) wavelengths. It has become clear that sensing using VIS-SWIR bands, based on the surface reflectance of sunlight, would benefit from complementary measurements using different technologies. Thermal infrared (TIR) sensing shows potential as a novel method for monitoring macro plastic litter floating on the water surface, as the physics behind surface-leaving TIR is different. We assessed a thermal radiance model for floating plastic litter using a small UAV-grade FLIR Vue Pro R 640 thermal camera by flying it over controlled floating plastic litter targets during the day and night and in different seasons. Experiments in the laboratory supported the field measurements. We investigated the effects of environmental conditions, such as temperatures, light intensity, the presence of clouds, and biofouling. TIR sensing could complement observations from VIS, NIR, and SWIR in several valuable ways. For example, TIR sensing could be used for monitoring during the night, to detect plastics invisible to VIS-SWIR, to discriminate whitecaps from marine litter, and to detect litter pollution over clear, shallow waters. In this study, we have shown the previously unconfirmed potential of using TIR sensing for monitoring floating plastic litter.

**Keywords:** plastic litter; thermal infrared; natural waters; pollution; UAV

## 1. Introduction

### 1.1. Current Research in the Remote Sensing of Floating Plastic Litter

There is growing global concern over the chemical, biological and ecological impacts of marine plastic pollution. Every year, 5 to 13 million metric tons of plastic litter enter the oceans [1], and exactly what happens to this plastic is unknown. Significantly less material has been observed on the ocean surface than has been predicted by budgeting exercises [2–4]. One of the reasons for the "missing sink" is the scarcity of data for most of the world's oceans. Remote sensing has the potential to provide long-term and spatially and temporally coherent observations on a global scale; remote sensing for plastics in the ocean has been evolving rapidly in recent years. To date, most progress has been made in the remote sensing of plastic macro litter (>5 mm) in the visible (VIS), near-infrared (NIR), and short-wave infrared (SWIR) wavelengths. Spectral light reflectance measurements of floating plastic macro litter have been made in situ [5,6], from unmanned aerial vehicles (UAVs) [6–8], airplanes [6,9], and the Sentinel-2 satellite [7,8,10,11]. Another recent approach is imaging floating plastic litter with VIS and applying automated image recognition techniques using an RGB (red-green-blue) camera mounted on a bridge [12], vessel [13,14], or aircraft [15]. These remote-sensing methods operate in the VIS-SWIR spectrum; measurements in other parts of the electromagnetic spectrum (using passive as well as active sensors) could improve remote-sensing algorithms for marine plastic

litter [16–18]. In this paper, we explore thermal infrared (TIR) sensing as one of these technologies [19,20], using a thermal imaging camera [7].

*1.2. TIR Remote Sensing*

The atmospheric windows in TIR are in the mid-wave infrared (MWIR, 3–5 μm) and long-wave infrared (LWIR, 8–14 μm) ranges. In these windows, TIR sensing could, for example, complement VIS-SWIR measurement for clear (dark-colored) plastic materials that are transparent (dark) in the VIS-SWIR spectrum but are opaque (bright) in the TIR spectrum. The physics behind surface-leaving radiance in TIR is not the same as in VIS-SWIR because the radiance in TIR is composed of reflected and emitted energy, while in VIS-SWIR, it is only reflected. Emissivity (and, hence, reflectivity) affects surface-leaving TIR, as does the surface temperature. According to Hecker et al. [21], hydrocarbon-based materials have a low specific heat capacity (the energy needed to raise the temperature of a material by 1 °C per unit mass of material), meaning that their temperatures change quickly compared to water, which has a high specific heat capacity. The resulting temperature differences should be detectable in TIR images. Sunlight can interact with plastic surfaces differently from water. Every object reflects a certain amount of light while absorbing the rest as heat energy; this energy is then re-radiated as TIR radiance. Dark-colored surfaces absorb more light and become warmer than light-colored surfaces. Clear plastic containers can trap TIR radiance inside them and warm up (greenhouse effect). Unlike remote sensing in the VIS-SWIR spectrum, TIR sensing requires no external illumination since the sensor records energy directly from the object. TIR sensing can, therefore, operate during both day and night.

TIR radiance is routinely measured in the thermal mapping of the Earth's surface, including the sea and other water surfaces, but other applications are possible, such as oil spill sensing in LWIR [22]. Garaba et al. [20] report the laboratory measurements of hyperspectral TIR hemispherical reflectance spectra (6–14.5 μm at 25 nm spectral resolution) for different beach-collected plastic litter items and natural items such as sand, shells, and algae. The data are freely available via an online repository, the PANGAEA database of the World Data Center for Marine Environmental Sciences [23]. Topouzelis et al. [7] imaged three 100 m$^2$ targets consisting of plastic bottles, bags and fishing nets floating in the Aegean Sea during the day (7 June 2018), using a UAV TIR camera 100 m above sea level. The plastic bottles and the plastic bags were identified with some degree of uncertainty as brighter surfaces.

The method for TIR remote sensing of floating plastic litter, which was proposed by Goddijn-Murphy and Williamson [19], is based on the different emissivity values and surface temperatures of water and of plastic and is explained in more detail in Section 1.3. Emissivity is the ratio of the energy radiated from a surface, as well as that radiated from a blackbody at the same temperature. The emissivity of water is very high, near one, and that of plastics is generally lower. The warmer an object, the more radiance is emitted in TIR, increasing with increasing emissivity. Therefore, for water and plastic at the same temperature, the emitted TIR radiance will be higher for water. However, water and plastic surfaces not only emit their own TIR radiance but also reflect TIR radiance from their surroundings. For an object of lower emissivity, the reflectivity is higher, which implies that plastic is generally a stronger reflector than water. Water readings will, therefore, more closely indicate the actual temperature of the water, while plastic will more closely indicate the temperature of the surroundings. In conclusion, the TIR signal of plastic is expected to improve with a lower value of plastic emissivity, enhanced by an increasing temperature difference between the air and the sea.

This dependence on air and sea temperatures complicates interpretations of the TIR signal, but this could also be used in our favor, for example, by using the difference between day and night measurements to detect the presence of plastic. An advantage of nighttime surveys, especially pre-dawn surveys, is that the uneven solar heating that occurs during the daytime is much reduced [24], making it easier to extract anomalous

pixels not related to solar heating. It is clear that we need to separate the temperature and emissivity dependencies to correctly interpret the TIR signal. Goddijn-Murphy and Williamson [19] apply a thermal radiative transfer model to floating plastic that addresses these dependencies, as described in Section 1.3. This model will also support remote-sensing algorithms based on the spectral signature of plastic in the TIR spectrum [20], as we expect it to apply to broad- as well as narrow-band TIR.

The goal of our field and laboratory experiments using a UAV TIR camera was to explore the following questions about the remote sensing of marine plastic litter in TIR. Can a UAV TIR camera operate as a TIR sensor? How does the TIR radiance transfer model perform? When and where can we expect the best results? Which plastic litter items give the best results? Can we separate plastic litter from other features on the water's surface? Can we see plastic litter on different surfaces other than water? Finally, how can TIR sensing complement VIS-SWIR sensing?

*1.3. Thermal Radiance Transfer Model*

The following is a short review of the radiance transfer model used by Goddijn-Murphy and Williamson [19]. Following Kuenzer and Dech [24], TIR radiance in a spectral band, captured by a sensor viewing an object, $L_{b,obj}$, is estimated as follows:

$$L_{b,obj} = \varepsilon_{obj}\tau_{atm}\cdot L_b\left(T_{obj}\right) + \left(1 - \varepsilon_{obj}\right)\tau_{atm}L_b(T_{sur}) + L_{b,path}. \tag{1}$$

The object can represent either a water body or plastic litter. In Equation (1), the first term represents the TIR radiance emitted by an object of temperature $T_{obj}$, the second term, TIR radiance of the surroundings, $L_{b,sur}$, which is reflected by the object, and the third term, TIR radiance reaching the sensor and having never interacted with the object. For an open ocean, $L_{b,path}$ has been approximated by $(1 - \tau_{atm})L_b(T_{atm})$, calculated as the emitted TIR radiance resulting from the absorption of radiation in the atmosphere, with $\tau_{atm}$ and $T_{atm}$ representing atmospheric transmissivity and temperature, respectively [25]. It is difficult to separate $L_{b,path}$ and $L_{b,sur}$ in practice, and we refer to both when we mention background radiance, $L_{b,back}$.

In Equation (1), $L_b(T)$ is the band radiance of a blackbody at temperature $T$, estimated from the spectral thermal radiance, $L$, of a blackbody. For temperature, $T$ (K), and wavelength, $\lambda$ (μm), $L$ increases with increasing temperature, according to Planck's law (Equation (2)):

$$L(\lambda, \mathrm{T}) = \frac{2hc^2}{\lambda^5}\left(\frac{1}{e^{hc/\lambda k_B T} - 1}\right)10^{24}\left[\mathrm{W\,m^{-2}sr^{-1}\mu m^{-1}}\right], \tag{2}$$

with $h$ representing Planck's constant ($6.626 \times 10^{-34}$ J s), $c$ representing the speed of light (299,792,458 m s$^{-1}$), and $k_B$ representing the Boltzmann constant ($1.3806 \times 10^{-23}$ J K$^{-1}$). Spectral radiance, $L(\lambda, T)$, is defined so that $L(\lambda, T)\mathrm{d}\lambda$ is the radiance within the intervals $\lambda$ and $\lambda + \mathrm{d}\lambda$. Thus, for a spectral band we calculate band radiance, $L_b$, from $\lambda_1$ to $\lambda_2$:

$$L_b(T) = \int_{\lambda1}^{\lambda2} L(\lambda, T)\mathrm{d}\lambda \left[\mathrm{W\,m^{-2}sr^{-1}}\right]. \tag{3}$$

Emissivity, $\varepsilon(\lambda)$, is the ratio of energy radiated from a surface at a given temperature, $M_b(T)$ (W m$^{-2}$), and that radiated from a blackbody (a perfect emitter) at the same *kinetic* temperature:

$$\varepsilon(\lambda) = \frac{M_{b,obj}(\lambda, T)}{M_b(\lambda, T)}. \tag{4}$$

All the TIR energy absorbed by a body in thermal equilibrium with its surroundings is emitted again so that $\varepsilon(\lambda)$ equals absorptivity, $\alpha(\lambda)$, the fraction of TIR energy absorbed (Kirchhoff's law). A blackbody absorbs all incident energy and, therefore, re-radiates 100% of the incident radiance. Most radiation sources are not blackbodies, however, as

some of the incident energy upon them may be reflected or transmitted. For all bodies (at thermal equilibrium), the conservation of energy dictates that the sum of absorptivity, reflectivity ($\rho$), and transmissivity ($\tau$), equals one (with $\rho/\tau$ representing the fraction of radiance reflected/transmitted). Radiation sources can be of different classes. For a blackbody, thermal radiance is described by Plank's curve ($\varepsilon = 1$). For a graybody, the curve is proportional to Planck's curve for all wavelengths ($\varepsilon < 1$). For a selective radiator, spectral radiant emittance varies not only with temperature but also with wavelength ($\varepsilon(\lambda) < 1$) [26]. Plastics and other man-made materials, as well as natural materials, are selective radiators, showing specific absorption features corresponding to their chemical composition. Garaba et al. [20] present hyperspectral TIR (6–14.5 μm) reflectance measurements that are collected from natural and anthropogenic material, including sands, shells, algae, nautical ropes, Styrofoam$^{\circledR}$, gunny sacks, and several fragments of plastic-based items.

We treat seawater and plastic as opaque ($\tau = 0$) and, together with $\varepsilon(\lambda) = \alpha(\lambda)$, we derive:

$$\varepsilon(\lambda) + \rho(\lambda) = 1. \tag{5}$$

We can consider seawater to be opaque, as water strongly absorbs TIR radiance. We assume plastic litter is also opaque in TIR, but this is not certain for all types. The TIR emission by transparent materials is reviewed by Gardon [27]. Assuming that the emitted and reflected TIR energy of an object (that object being water or plastic) are Lambertian, meaning that the emitted TIR radiance is the same in all directions and, hence, $M_b(T) = \pi L_b$, we calculated the total captured by a TIR sensor, $L_{b,obj}$, using Equation (1).

The emissivity of water, $\varepsilon_w$, and of aluminum foil, $\varepsilon_{alu}$, is 0.98 and 0.036, respectively, for a wavelength band of 8–14 μm [24], and that of plastic materials, somewhere in between [19]. Looking at Equation (1), this means that the radiance of water/aluminum foil in TIR is mostly emitted/reflected. We used water as an approximation of a blackbody and aluminum foil to assess the background radiance. We applied the above theory to a broad band in LWIR, but the theory should also apply to narrow bands and to other TIR wavelengths, such as MWIR.

## 2. Materials and Methods

The focus of this paper is the TIR imaging of floating plastic litter at sea from a UAV, using a FLIR (forward-looking infrared) camera. The UAV also carried NIR and RGB cameras that were taking concurrent recordings, to see how TIR imaging could complement VIS-NIR data. We supported our field measurements with FLIR measurements in the laboratory, and we used Equations (1)–(3) to estimate the captured TIR radiance and evaluate FLIR camera response.

### 2.1. FLIR Camera and Image Processing

We used the FLIR Vue Pro R 640 (FLIR, Wilsonville, OR, USA) for imaging in the spectral band, at 7.5–13.5 μm (wavenumber 1333–741 cm$^{-1}$) in LWIR. "R" stands for radiometric meaning reading the intensity of thermal radiation. The specifications of this camera are: sensor resolution = 640 × 512 px, field of view (FOV) = 32°, and focal length = 19 mm. The ground sample distance (GSD) at 30 m of altitude was 2.7 cm/px, with a frame height and width of 13.9 and 17.4 m, respectively. The FLIR camera was not calibrated by the manufacturer.

We recorded FLIR images as a 14-bit TIFF (tag image file format), producing uncompressed and unprocessed grayscale images (raw), making the FLIR's radiometric settings irrelevant. Each TIFF file was imported into MATLAB and, for each image, we selected the surface of interest (water, aluminum, plastic, or wood) and calculated the average digital numbers (DN) of all pixels in the selection. Kelly et al. [28] recommend selecting at least 10 pixels; our selections were at least 20 pixels for 30-meter-altitude UAV images and were >1000 in the laboratory. We defined delta as the DN difference of the target surface

pixels and water surface pixels, to evaluate the TIR signal from plastic floating on water, as follows:

$$\text{delta(target)} = \text{DN(target)} - \text{DN(water)}. \tag{6}$$

In the field, we used 1-min averages of 1-Hz measurements (DN and delta) following the baseline surface radiation network (BSRN) [29]. On the recommendation of Kelly et al. [28], the FLIR camera was powered up at least 30 min before launch, and its recording minute was taken after the UAV had been in position above the targets for at least five minutes (to maximize the stabilization of the FLIR camera). In the laboratory, where environmental conditions were stable, we used a single image. The FLIR camera in the laboratory was powered up 1 h before measurements were taken. The FLIR images showed vignetting, i.e., a bright center and darker areas around the edges [28]. For deriving the delta value, we, therefore, retrieved the DN values from the target surface and the nearby water surface. It was necessary to achieve "flat" images for stitching the UAV images together in a mosaic, to prevent vignetting artifacts in the mosaic, as seen in [7] (Figure 2e). We applied the MATLAB function imflatfield, with a sigma of 30, to remove the vignetting patterns; this could produce image artifacts. Section 3.4 details the measuring of the consequences of the flatfield correction for DN and delta. In the laboratory, where we could place the target in the center of the view of the FLIR, we did not apply the flatfield correction. Hence, we used both flatfield corrected and uncorrected images in our analysis of the TIR signal, as follows:

i.    Flatfield, corrected for comparing the DN values of targets in UAV images;
ii.   Uncorrected for deriving delta in UAV images, with DN(water) taken close to DN(target);
iii.  Uncorrected for DN(water) in the UAV images, taken from the center of the images;
iv.   Uncorrected for FLIR images taken in the laboratory, with targets in the center of the view.

The corresponding workflow and their results are illustrated in Figure 1.

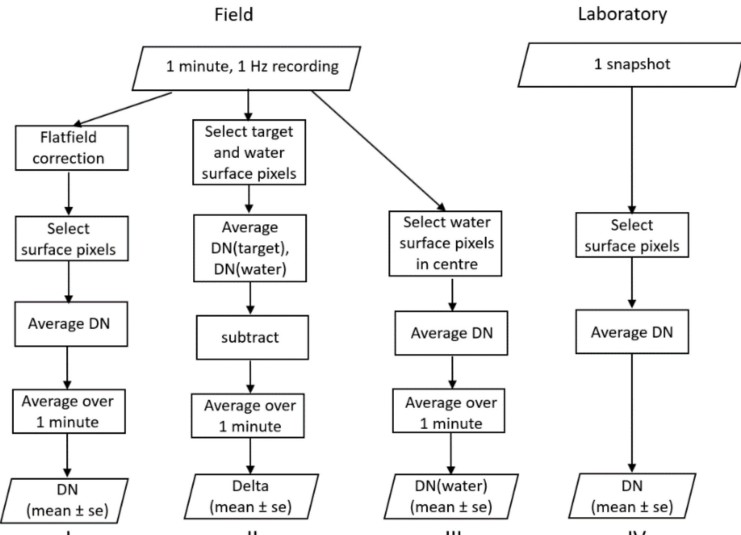

**Figure 1.** Diagram showing the image-processing workflows for FLIR field measurements (I–III) and laboratory measurements (IV). Processing workflows I–III before the 1-min average applies to individual images; se is standard error of the mean. Surface pixels in I and IV can be of the water or the target.

In the following, we report the DN and delta values, specifying whether the flatfield correction was applied.

*2.2. NIR and RGB Cameras*

We used a MAPIR 3N (San Diego, CA, USA), a NIR camera with a single band at 850 nm. Specifications of this camera are: sensor resolution = 4000 × 3000 px, FOV = 41°, focal length = 8.25 mm. The GSD of the MAPIR at 30 m altitude was 0.564 cm/px, with a frame height and width of 17.1 and 22.7 m, respectively. The RGB camera model was a ZENMUSE X5 (DJI, Shenzhen, China) in the UAV housing, covered by a dome to protect it from sand and spray. The specifications of the ZENMUSE X5 are: sensor resolution = 4608 × 3456 px, FOV = 71°, focal length = 15 mm. The GSD at 30 m of altitude was 0.75 cm/px, with a frame height and width of 26.9 and 34.6 m, respectively. The ZENMUSE X5 camera lens was set to AF (auto-focus) mode. In this paper, we show JPEG images of the MAPIR and ZENMUSE X5 to illustrate their views during the daytime surveys. The FLIR, NIR, and RGB cameras were fixed at different positions on the UAV, which explains why their images did not line up perfectly.

*2.3. Temperature, Light Intensity, and Humidity*

2.3.1. Measurements

We need true kinetic temperature measurements of the water and target surfaces, as well as their surroundings if we want to understand the TIR signal. Thermal inertia, defined as the resistance of a material to heating, is the product of the specific heat capacity, density, and thermal conductivity of the material. Man-made materials (e.g., metals and hydrocarbon-based materials) do not easily remain at a constant temperature, due to a low specific heat capacity [21]. In the field, therefore, it is important to measure temperatures at the exact moment of capturing the FLIR image. Temperatures can change quickly, for example, when targets move in and out of sunlight due to variable cloud cover. Water has a high specific heat capacity and its temperature, $T_w$, does not change as quickly as the temperature of targets. Another challenge is the requirement to measure temperature at the surface of the material. For a water body, we can measure a vertical temperature profile and extrapolate this for the surface temperature. For plastic and metal surfaces, it is difficult to establish the true kinetic surface temperature because a contact thermometer influences the surface temperature it measures, due to interferences such as reducing cooling by airflow and heating by the absorption of light. Although measuring air temperature, $T_{air}$, may seem straightforward in comparison because adequate immersion is almost guaranteed, it is not that easy. Air and gases are not efficient heat-transfer media; even a modest heat input to the sensor can cause the readings to be high [30]. The thermometer should, therefore, be protected from airflow, (sun)light, and other sources of radiation.

Handheld probe and data-logging thermometers were used to measure the kinetic temperatures of air, water, and the floating surfaces, and dataloggers were used to record light intensity (*I*) and relative humidity (RH) (details of the sensors are given in Section 2.3.2). In the laboratory, we mostly used handheld sensors, while in the field, dataloggers with handheld sensors were used as a backup. In the field, dataloggers took measurements every 10 s (we found that the 1-minaverages of the 0.1 Hz data were not significantly different from the 1 Hz data) and we smoothed the data using a 1-min window. We matched the FLIR-imaging minute with the temperature-recording minute, then we took handheld measurements as close to the FLIR recording as possible. We will assess the performance of the thermometers in Section 3.3. We could not ascertain which measurement was most accurate in the case of handheld devices or dataloggers; we will inform the reader what types were used.

2.3.2. Sensor Details

**Handheld sensors.** A handheld type-K air/gas probe measured the ambient air temperature, a type-K right-angle (RA) surface probe measured the temperature of the target surfaces (plastic and aluminum), and a waterproof Pt100 sensor on a cable measured the water temperature profiles (Thermosense, Bucks, UK). The accuracy of the type-K

readings was $\pm$ 0.15% of reading +1 °C, and of the Pt100 sensor, $\pm$ 0.05% of reading +0.5 °C. We measured the temperatures as close in time to the FLIR imaging as possible.

**Dataloggers.** We used the iButton® DS1922L Thermochron data loggers ($-40$ °C to +85 °C) to log the temperatures, with 8-bit (0.5 °C) resolution, an iButton® DS1923 Hygrochron temperature/humidity logger (RH, with 8-bit (0.6%) resolution) (MSL, Newbury, UK), and a HOBO® (Onset, Bourne, MA, USA) miniature temperature and light logger (temperature $\pm$ 0.53 °C, from 0° to 50 °C) and with a light sensor designed for the measurement of relative light levels from 0 to 320,000 lux (0 to 30,000 lumens/ft$^2$). An iButton datalogger, submerged in a waterproof casing, measured the water temperature approximately 15 cm below the water surface. We measured the surface temperatures of the targets by placing iButtons in miniature plastic bags, tightly taped to the surface. In the field, we measured $T_{air,2m}$ with iButton and HOBO loggers that were attached to a small, expanded polystyrene platform on a stick approximately 2 m above the sand, and $T_{air,30m}$ using an iButton attached to the UAV. We tried to place the dataloggers out of the direct sun, wind, and drafts, but this was not always possible, especially not in the case of those that were on the target surfaces. In Section 3.3.1, we evaluate the systematic errors between the handheld devices and dataloggers. We estimated room temperature in the laboratory by attaching an iButton to the FLIR camera tripod, approximately 1 m off the ground.

*2.4. UAV Surveys*

The Tetra TD-7 UAV (Tetra, Lincoln, UK) had a ~28-min flight time when unladen. It carried the three downward-looking cameras in a nadir view to take concurrent images. The objective of a flight was to image floating plastic litter targets and the aluminum foil reference target at sea, by hovering over them in one position at a 30 m altitude. We operated the UAV using the DJI GO app, by which we could follow the view of the RGB camera and position the UAV. For the night surveys, we deployed two green Lumica 30-min glow sticks, fixed to moorings in the target area, to help find our targets. The floating glow ticks did not appear in the FLIR images. Our survey site was in Thurso Bay in the Pentland Firth, on the north coast of Scotland (58.5987°N, 3.5166°W), where we performed four UAV surveys during the day and night and in different seasons (Table 1). The ambient temperatures during the surveys were sufficiently different to cover a range of conditions. During surveys 1, 2, 3, and 4, all floating targets were deployed at sea 20, 45, 30, and 50 min before the FLIR recording minute, respectively.

**Table 1.** Details of four field surveys, LT = UTC + 1; location: Thurso Bay (58.5987°N, 3.5166°W).

| Survey | Day (2021) | | LT | Sky Condition | Sea State |
|---|---|---|---|---|---|
| 1 | 1 April | day | 07:40 | Cloudy | smooth |
| 2 | 23 April | night | 04:14 | Overcast (no stars) | slight |
| 3 | 3 August | day | 12:01 | Overcast (100% cloud cover) | calm (rippled) |
| 4 | 4 August | night | 01:41 | Clear sky (stars and red moon) | calm (smooth) |

During the field surveys, we flew the UAV over 0.5 × 0.5 m targets floating at sea, connected to moorings with a 1.5-meter rope. We created these targets using plastic litter items composed of expanded polystyrene (EPS), polyethylene (PE), polyethylene terephthalate (PET), high-density polyethylene (HDPE), and low-density polyethylene (LDPE). Each target consisted of one kind of litter item; these were numbered 1–7, as follows:

1. PET bottles, clear (0.5 L);
2. PET bottles, clear (2 L);
3. EPS foam board, white (thickness 5 cm);
4. EPS foam board, blue (thickness 3 cm);
5. HDPE milk bottles, semi-transparent, white (2.3 L);
6. LDPE/HDPE binbag, black, two thin layers;
7. PE tarpaulin, white, single-layer;
8. Aluminum foil (wrapped around 3);

9.      Wooden tree trunk disk (thickness 4 cm, radius 29 cm).

Polymer compositions of items 6 and 7 were not specifically known; therefore, we refer to these as a binbag and tarpaulin, respectively. Number 8 was a reference target used to estimate the background TIR radiance. We deployed the wood disk (9) during the summer surveys for comparison with plastic. Targets 1, 2, 3, and 5 were previously used by Goddijn-Murphy and Dufaur in their spectral light reflectance measurements [5].

### 2.5. Atmospheric Parameters from ERA5

For $T_{air}$ at higher altitudes, we obtained the temperature profiles from the data set "ERA5 hourly data on pressure levels, from 1979 to present" from the Climate Data Store (CDS). The CDS provides hourly reanalysis calculations on a $0.25° \times 0.25°$ grid for 37 pressure levels, from 1000 hPa to 1 hPa (111 m to 32,435 m) [31]. We calculated the altitude in meters from the pressure level in hPa, following the NOAA's National Weather Service [32]. We also downloaded $0.25° \times 0.25°$-resolution atmospheric data from "ERA5 hourly data on single levels, from 1979 to present" [33]; these were STRD (surface thermal radiation downward) values at the Earth's surface, including clouds, LCC (low cloud cover), and CBH (cloud base height). We interpolated these ERA5 data between 3–4°W and 58–59°N for the survey location and time (Table 2).

**Table 2.** ERA5 atmospheric parameters from CDS interpolated to the survey location and time. STRD (surface thermal radiation downward) is measured at the Earth's surface and includes clouds; LCC (low cloud cover); CBH (cloud base height).

| Survey | Tair, 111 m ($°C$) | STRD ($10^6$ J/m$^2$) | LCC (0–1) | CBH km |
|--------|---------|------|-----|--------|
| 1 | 1.9 | 1.0395 | 0.81 | 0.9231 |
| 2 | 5.0 | 1.1465 | 0.79 | 0.5344 |
| 3 | 14.6 | 1.2696 | 0.82 | 0.9115 |
| 4 | 13.7 | 1.1582 | 0.30 | 1.5682 |

### 2.6. FLIR Measurements in the Laboratory

We turned off all the lights and heating radiators in the room and closed the window blind to reduce the incoming daylight. For each type of plastic litter, the measurement (workflow IV in Figure 1) was repeated three times, using different items of the same type, while we used single measurements of aluminum reference ($0.1 \times 0.1$ m version of target 8 in Section 2.4) and the water surface on both sides of the three measurements. The FLIR camera was mounted on a tripod, with its legs against a 1 m $\times$ 1 m square of 5 cm-thick EPS foam board on the floor, on which we placed the objects we imaged. These comprised a basin filled with water and floating targets. We used metal pliers to handle the targets, as contact with bare fingers would leave warm fingerprints. The camera was tilted downward at an angle of approximately 45° so that it did not record its own reflected heat. We positioned all targets in the center of the FLIR view and did not need to apply a flat-field correction to rectify the vignetting of the camera. We used water as a near-perfect TIR emitter (black body) and aluminum as a near-perfect TIR reflector. Water temperature was regulated by using hot water from the tap and ice from the freezer. The aluminum foil was wrapped around an oven brick and placed in a furnace to achieve temperatures of > 100 °C. While imaging the hot brick, it was placed on a heat-resistant tile.

In the laboratory experiments, we used the same plastic litter items (dry and wet) as in the field surveys. The laboratory experiments comprised: measuring the FLIR camera response to water temperature and calculated TIR radiance; measuring the FLIR camera response to aluminum foil temperature; measuring the FLIR camera response to floating plastic litter and as a function of water temperature and plastic litter density; and a bio-fouling experiment.

*2.7. Biofouling Experiment*

Biofouling can affect TIR sensing in different ways. Biofouling can cause plastic litter to sink below the water surface and remove it from view, but this problem is beyond the scope of our research, and we refer to other studies on this subject (e.g., [34]). Biofouling can produce a surfactant film on the water surface, which may be seen by a thermal imaging camera [22]. This is also beyond the scope of this project. The aim of our experiment was to assess how biofouling on a plastic litter affected the surface-leaving TIR, as measured with the FLIR camera. This could be through biofouling changing (1) surface emissivity, (2) surface temperature, and (3) wetness of the surface.

In our biofouling experiment, we used two sets of a small PET soft-drink bottle (1), blue EPS foam board (4), an HDPE milk bottle (5), a white tarpaulin (6), and a black bin bag (7) (numbers of targets as listed in Section 2.4). Bottles were uncapped and the sizes of 4, 6, and 7 were 10 cm × 10 cm. In one set, we had three of each litter item, all pre-dried (at 30 °C for 24 h) and pre-weighed (using an Avery Berkel FA-114 electronic balance, Smethwick, UK). Following the procedure of Fazey and Ryan [34], we tethered a set of plastic litter items to a 1-meter PVC (polyvinyl chloride) pipe with 15-centimeter pieces of strong fishing line. Both racks were deployed on 13 April 2021 (expecting to catch the spring algal bloom) in a sheltered location under a floating pontoon in Scrabster harbor (58.6122°N, 3.5490°W). We also kept a clean, 'virgin', set for comparison. One rack was retrieved after two weeks to assess the first signs of biofouling and the other was retrieved after three months, for denser biofouling cover [34]. The biofouled items were re-dried and re-weighed after retrieval, to estimate the mass growth of biofouling. The biofouled items, both wet and after drying, were FLIR-imaged on an EPS panel wrapped in aluminum. For each wet (dry) plastic litter item, we measured a wet (dry) virgin item, followed by three wet (dry) biofouled items. We also imaged the TIR radiance leaving the water in a small container, along with an aluminum reference. Photos were taken in NIR and VIS.

## 3. Results

*3.1. Assessing the FLIR Camera Response*

We used FLIR images of the water surface to estimate the FLIR response because water emissivity, $\varepsilon_w$, is close to one, so that the water-leaving band radiance captured by the FLIR camera is mostly controlled by the TIR emittance of the water surface (Equation (1)). For measurements taken in the laboratory, the FLIR camera response, quantified by the DN of water, to $T_w$ was linear over $T_w$ ranges (5–35 °C and −9–1 °C) (Table 3).

**Table 3.** Curve fitting results of DN as a function of $T_w$ to $p_1 x + p_2$, derived in the laboratory, with $T_{air}$ and $T_w$ measured using handheld sensors. Std is standard deviation of the mean and RMSE root-mean-square-error of the fit.

| $T_{air}$ (mean ± std) (°C) | $T_w$ (°C) | $p_1$ | $p_2$ | $R^2$ | RMSE (DN) |
|---|---|---|---|---|---|
| 19.8 ± 0.2 | 6 to 35 | 24.9 ± 0.7 | 6852 ± 15 | 1.00 | 10 |
| 22 ± 1 | 4 to 35 | 22.9 ± 0.5 | 6922 ± 12 | 1.00 | 13 |
| 19.7 ± 0.3 | −9 to 1 | 31 ± 3 | 6820 ± 10 | 0.98 | 18 |

Over the whole $T_w$ range (−9 to 35 °C), a quadratic relationship was closer (Figure 2):

$$DN = -0.169 T_w{}^2 + 30.6 T_w + 6839 \tag{7}$$

where ($R^2$ = 0.99; RMSE = 29 DN), with uncertainty in the quadratic coefficient, linear coefficient, and offset being ± 0.06, ± 2, and ± 10, respectively.

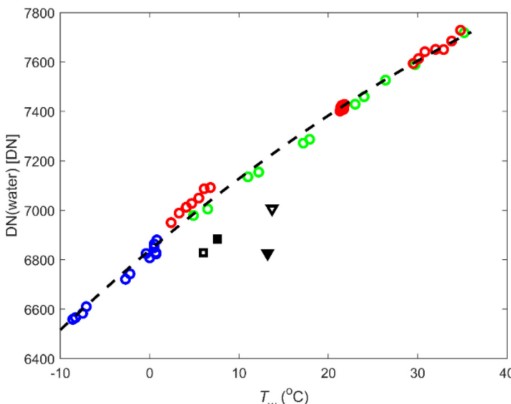

**Figure 2.** Scatter plot of the DN of the water surface (uncorrected) as a function of $T_w$ (°C), with circles indicating laboratory measurements in green $T_{air}/T_w = 19.8 \pm 0.2/6$–35 °C, red $22 \pm 2/4$–35 °C, and blue $19.5 \pm 0.5/-9$–1 °C; black squares/triangles indicate the April/August field surveys, with open/filled symbols representing day/night. The dashed line indicates Equation (7).

The inverse of Equation (7) (using kDN) is:

$$T_w = 9.7\text{kDN}^2 - 101\text{kDN} + 236 \qquad (8)$$

where ($R^2 = 0.99$; RMSE = 1), with uncertainty in the quadratic coefficient, linear coefficient, and offset being $\pm 3$, $\pm 40$, and $\pm 140$ respectively. Equation (8) was valid for indoors and $T_{air}$ of $21 \pm 2$ °C; in the field, the DN(water) was significantly lower.

We could calculate the first and second terms of Equation (1) using Equations (2) and (3), but not $L_{b,path}$. Therefore, we estimated $L_{b,w} - L_{b,path}$ for $T_w$ using $T_{sur} = T_{air}$, with $\varepsilon_w = 0.98$ and $\tau_{atm} = 1$. $T_{sur} = T_{air}$ was an approximation for both variables in the laboratory and over the open ocean. However, since we multiplied $L_b(T_{sur})$ with $(1 - \varepsilon_w)$, we deemed this to be acceptable.

A scatter plot of $L_{b,w} - L_{b,path}$ against DN illustrated an approximation of the captured band radiance leaving the water surface, as a function of DN (Figure 3). An exponential curve fit resulted in:

$$L_{b,obj} - L_{b,path} = 0.41 e^{0.65 \cdot 10^{-3} DN} \qquad (9)$$

where ($R^2 = 0.99$, RMSE = 0.8), with uncertainty in the exponential slope (intercept) of $\pm 0.02 \cdot 10^{-3}$ ($\pm 0.05$). Figure 3 shows how, for the same DN, the $L_{b,w} - L_{b,path}$ estimated for the field surveys was higher than in the laboratory; we explained this finding by $L_{b,path}$ being lower over the open ocean than indoors, where the walls and other objects radiated in TIR. We investigated this further using the aluminum foil reference in Section 3.2.

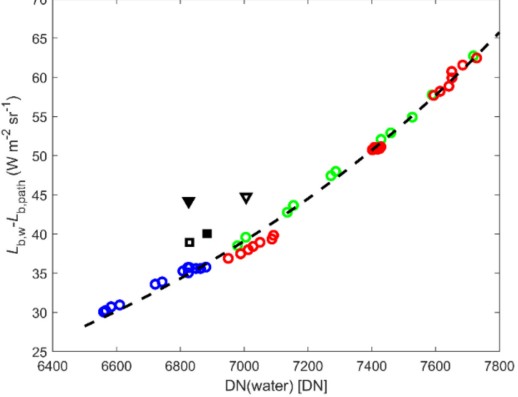

**Figure 3.** Scatter plot of the modeled $L_{b,w} - L_{b,path}$ (Equations (1)–(3)) as a function of DN, using data from Figure 2. The dashed line indicates Equation (9).

### 3.2. Response of the FLIR Camera to Background TIR Radiance

Aluminum foil-leaving radiance, $L_{b,alu}$, is dominated by the background TIR radiance and not by the aluminum foil temperature, $T_{alu}$, as its emissivity is near zero. It follows from Equation (1) (with $\tau_{atm} = 1$), that:

$$L_{b,alu} \approx L_b(T_{sur}) + L_{b,path} \tag{10}$$

We could recognize this in the scatter plot of DN a function of $T_{alu}$ obtained in the laboratory (Figure 4); DN was near-constant over the $T_{alu}$ range and slightly higher/lower for a higher/lower $T_{air}$. The estimated background TIR in the laboratory from a mean DN(alu) of $7.415 \pm 0.07$ kDN was related to $T_{air} = 20.4 \pm 3\,^{\circ}$C (Equation (8)). This was not significantly different from the mean measured $T_{air}$ of $21.5 \pm 1.5\,^{\circ}$C and implied that in approximation, the room, the walls, and everything inside was in thermal equilibrium. It also implied that the aluminum foil reference gave a reasonable estimate of $L_{b,back}$ and background temperature. In the field, DN(alu) was significantly lower and ranged from 5.7 to 6.7 kDN (Figure 4). If we applied Equation (8), these values were associated with background temperatures of $-25$ to $-4\,^{\circ}$C.

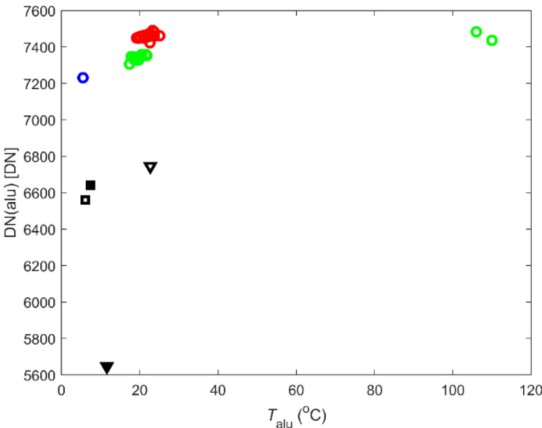

**Figure 4.** Scatter plot of DN on the aluminum foil surface (field survey data, corrected with sigma = 30) as a function of $T_{alu}$ (°C), with circles indicating laboratory measurements, in green $T_{air} = 19.8\,^{\circ}$C, red $T_{air} = 22\,^{\circ}$C, and blue $T_{air} = 19.3\,^{\circ}$C; black squares/triangles indicate April/August field surveys, with open/filled symbols representing day/night.

### 3.3. Temperatures

#### 3.3.1. Environmental Temperatures

In the laboratory, the surface and water temperatures measured using the handheld sensors and iButton dataloggers were similar, but datalogger $T_{air}$ was about 1 °C higher than $T_{air}$, measured using the handheld air/gas probe. In the field, the difference between $T_{air,2m}$ when measured using a logger and a handheld sensor was $\pm 2\,^{\circ}$C (Table 4); the dataloggers recorded a higher (lower) $T_{air,2m}$ during the day (night). The difference between the iButton and HOBO $T_{air,2m}$ was insignificant. The measured water temperature, $T_w$, with the iButton datalogger was up to 0.5 °C higher than that measured with the handheld sensor. We did not find evidence of thermal stratification of the water column using the latter.

**Table 4.** Environmental conditions, measured during the UAV survey using handheld probes and dataloggers, with temperatures in °C and wind speed in km/hr.

| Survey | Handheld | | | | Datalogger | | | | | |
| | $T_{air,2m}$ | $T_w$ | $T_{sand}$ | Wind | iButton $T_{air,2m}$ | HOBO $T_{air,2m}$ | iButton $T_{air,30m}$ | iButton $T_w$ | iButton RH% | HOBO $I$ (lux) |
|---|---|---|---|---|---|---|---|---|---|---|
| 1 | 5.7 | 5.6 | 6.0 | 5 | 6.1 | 5.9 | 7.6 | 6.1 | 66.8 | 5511 |
| 2 | 7.4 | 7.3 | 6.8 | 1 | 6.6 | 6.4 | 7.1 | 7.6 | 87.6 | 0 |
| 3 | 17.0 | 13.2 | 17.7 | 3–16 | 19.0 | 19.2 | 17.9 | 14.1 | 66.0 | 14,467 |
| 4 | 13.5 | 12.7 | 13.9 | 2–6 | 11.6 | x | 12.6 | 13.6 | 98.8 | 0 |

### 3.3.2. Surface Temperatures of Floating Plastic

In the field, we saw all temperatures vary by the minute while the targets were deployed at sea (Figure A4). It was, therefore, important to match temperatures to the FLIR recording minute (Table 5) for a valid comparison. We could relate the rising temperatures during the daytime surveys (Figure A4a,c) with increasing light intensity. During the summer night survey, all temperatures started dropping around 01:25 (Figure A4d). At the same time, RH started rising from 97% to 100%. In the following, we use the means over the FLIR recording minute (Table 5).

**Table 5.** Target surface, water, and air temperatures during the FLIR recording minute, using the iButton dataloggers (1-min averaged over 0.1 Hz measurements).

| Survey | Temperature (°C) | | | | | | | | |
| | Water | PET S | PET L | EPS White | EPS Blue | HDPE | Binbag | Tarp | Alu | Air, 2 m |
|---|---|---|---|---|---|---|---|---|---|---|
| 1 | 6.1 | 6.1 | 6.6 | 5.6 | 6 | 5.8 | 6.1 | 6.1 | 6.1 | 6.1 |
| 2 | 7.6 | 7.1 | 6.6 | 6.8 | 6.6 | 6.1 | 7.5 | 7.6 | 7.4 | 6.6 |
| 3 | 14.1 | 15.1 | 17.6 | 24.6 | 22.6 | 24.1 | 15.6 | 14.8 | 22.7 | 19.0 |
| 4 | 13.6 | 12.1 | 11.1 | 11.0 | 11.1 | 11.7 | 12.1 | 12.6 | 11.6 | 11.6 |

In survey 1, $T_{air,2m}$ was equal to $T_w$ and all target surfaces, except PET L, were at the same temperature or were colder than the water. In night survey 2/4), $T_{air,2m}$ was 1/2 °C lower than $T_w$, and all target surface temperatures were the 'same or lower'/lower than $T_w$. Measured temperatures during daytime survey 3 around noon were very different. $T_{air,2m}$ was approximately 5 °C higher than $T_w$, and all target surfaces were warmer than the water. The binbag was warmer than the tarpaulin, which was likely related to the binbag (tarpaulin) being black (white) and absorbing (reflecting) sunlight. The temperatures of EPS boards and HDPE bottles were the highest, being up to 10 °C higher than the water. As in survey 1, PET L was warmer than PET S, but not warmer than air. It may be that the bottles were not in the sun for long enough for the greenhouse effect to raise the surface temperature.

Indoors, in the absence of wind and sunlight, when $T_{air}$ was 19.8 °C and $T_w$ ranged between 4.9 and 35.5 °C, we observed surface temperatures of three (two)-dimensional litter items that were closer to air (water). For two-dimensional items, wet and dry surface temperatures were not significantly different, but for the three-dimensional items, wetness brought the temperature generally closer to $T_w$. In the field, we recognized the two-dimensional binbag and tarpaulin and three-dimensional PET S, which were closer to the water temperature than the other three-dimensional targets.

### 3.4. FLIR Signals of Floating Plastic

Conditions during the surveys are presented in Tables 1, 2 and 4, and the surface temperatures in the field are given in Table 5. Snapshots of the images taken with the three cameras of the floating plastic targets at sea are shown in Appendix A.1.

Except during survey 3, all targets looked cooler in the images than the water, while the aluminum reference and EPS white looked cooler than the water in all surveys (Figures A1–A3).

Figures A5 and A6 illustrates the DN/delta values of all the targets and water during the FLIR recording minute; we could see how they varied and followed each other during this minute. The variation was smallest during survey 4, under a clear night sky. Table 6 presents all DN- and delta values retrieved during the UAV surveys. Table 6b lists delta calculated using flatfield-corrected images (relating to the DN values in Table 6a), and Table 6c, delta values calculated using uncorrected images. A comparison revealed that the flatfield correction reduced delta, implying that it brought the DN values of different targets closer together. Data from Table 6a,c, except for aluminum foil, are illustrated in the form of bar charts (Figure 5). Figure 5b shows that for surveys 1, 2, and 4, |delta| was largest for the EPS boards (3 and 4), followed by the bottles (1, 2, and 5) and the binbag (6). We found the same ranking in the laboratory for wet and dry surfaces ($T_w > T_{air}$) and dry surfaces ($T_w < T_{air}$). We did not expect a large |delta| value for the tarpaulin (7) in survey 4 (the only survey with useful tarpaulin images) from the laboratory experiments. Survey 3 was different, as the sun warmed all targets; this likely explained the large positive delta for the black binbag (6). We could not relate delta < 0 for the white EPS board (3) to a low surface temperature as it was the warmest measured surface (Table 5). For all four surveys, white EPS revealed the lowest DN levels of all targets (except aluminum), including blue EPS.

It is difficult to keep the target surfaces dry or wet when deployed at sea; hence, we studied the effect of wetness in the controlled environment of the laboratory. These experiments showed that wetting the surface decreased the sensitivity of delta to the presence of plastic on the water surface, except when $T_w \approx T_{air}$. When $T_w \approx T_{air}$, delta ≈ 0 for dry surfaces and delta < 0 for the wet surfaces; this was presumably caused by evaporation cooling the surface. In the field, the FLIR signal did not correlate with surface temperature.

Figure 5a shows that DN was highest during survey 3; this can be explained by the sun warming all surfaces (Table 5). This resulted in different delta responses from those retrieved during the other surveys (Figure 5b) and in the laboratory. From the theory and the laboratory experiments, we expected delta values of near-zero for $T_w \approx T_{air}$, but for surveys 1, 2, and 4 (when $T_w \approx T_{air,2m}$ when averaging the handheld and datalogger measurements), the delta value was significantly smaller than zero for all targets. Delta was most negative for survey 4, when we took measurements under a clear night sky.

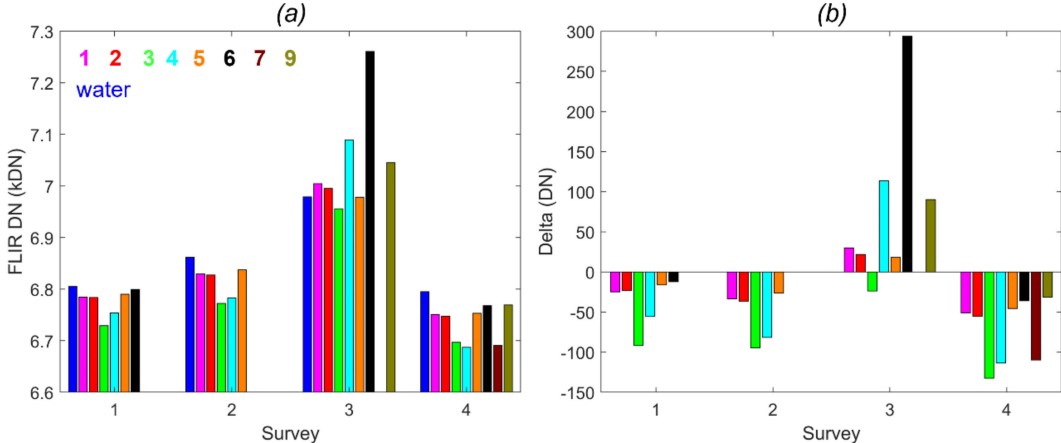

**Figure 5.** (**a**) FLIR signal (kDN) for all imaged surfaces during all UAV surveys, except the aluminum foil reference (flatfield-corrected with sigma = 30), and (**b**) delta (DN) using uncorrected images. Legends indicate: 1-PET S, 2-PET L, 3-EPS white, 4-EPS blue, 5-HDPE, 6-bin bag, 7-tarpaulin, and 9-wood. Error bars were too small to show.

**Table 6.** (**a**) FLIR signal (kDN) for all imaged surfaces, obtained from survey "S". Values are the means over 60 images (**a**) after flatfield correction with sigma = 30, SE < 0.001 kDN (**b**) concurring delta (DN), SE < 0.1 DN, and (**c**) delta (DN), estimated using uncorrected FLIR images, SE < 0.1 DN.

| (a) | | | | | | | | | |
|---|---|---|---|---|---|---|---|---|---|
| **FLIR kDN** | | | | | | | | | |
| **S** | **Water** | **PET S** | **PET L** | **EPS White** | **EPS Blue** | **HDPE** | **Binbag** | **Tarpaulin** | **Alu** | **Wood** |
| 1 | 6.805 | 6.785 | 6.784 | 6.729 | 6.754 | 6.79 | 6.799 | NaN | 6.56 | NaN |
| 2 | 6.862 | 6.83 | 6.827 | 6.772 | 6.783 | 6.837 | NaN | NaN | 6.641 | NaN |
| 3 | 6.979 | 7.004 | 6.995 | 6.956 | 7.089 | 6.978 | 7.261 | NaN | 6.745 | 7.045 |
| 4 | 6.795 | 6.751 | 6.747 | 6.697 | 6.687 | 6.753 | 6.768 | 6.691 | 5.647 | 6.769 |

| (b) | | | | | | | | | |
|---|---|---|---|---|---|---|---|---|---|
| **Delta (DN)** | | | | | | | | | |
| **S** | **PET S** | **PET L** | **EPS White** | **EPS Blue** | **HDPE** | **Binbag** | **Tarpaulin** | **Alu** | **Wood** |
| 1 | −21 | −20 | −71 | −48 | −14 | −9 | NaN | −255 | NaN |
| 2 | −29 | −31 | −85 | −73 | −23 | NaN | NaN | −191 | NaN |
| 3 | 23 | 14 | −21 | 102 | 14 | 265 | NaN | −373 | 85 |
| 4 | −41 | −44 | −105 | −96 | −38 | −24 | −96 | −1073 | −26 |

| (c) | | | | | | | | | |
|---|---|---|---|---|---|---|---|---|---|
| **Delta (DN)** | | | | | | | | | |
| **S** | **PET S** | **PET L** | **EPS White** | **EPS Blue** | **HDPE** | **Binbag** | **Tarpaulin** | **Alu** | **Wood** |
| 1 | −25 | −23 | −92 | −56 | −16 | −12 | NaN | −262 | NaN |
| 2 | −34 | −37 | −95 | −82 | −26 | NaN | NaN | −219 | NaN |
| 3 | 30 | 22 | −24 | 114 | 18 | 294 | NaN | −245 | 90 |
| 4 | −51 | −55 | −133 | −114 | −46 | −36 | −110 | −1214 | −31 |

### 3.5. Background TIR Radiance over the Open Ocean

In Section 3.2, we measured the response of the FLIR camera to background TIR in the laboratory and compared these with observations of DN(alu) during our four field surveys. Over the open ocean, background TIR radiance appeared to be generated in colder, sub-zero temperature environments, i.e., at higher altitudes than near the surface. This was best seen in summer day survey 3, when $T_w$ was 14.1 °C and $T_{air}$ at 2, 30, and 111 m and 19.0, 17.9, and 14.6 °C, respectively, but where more TIR radiance left the water than the background TIR radiance reflected from aluminum foil. We measured DN(alu) increasing with the increasing LCC, lowering the CBH estimated from ERA5 (Figure 6); a relationship with ERA5′s STRD was not found.

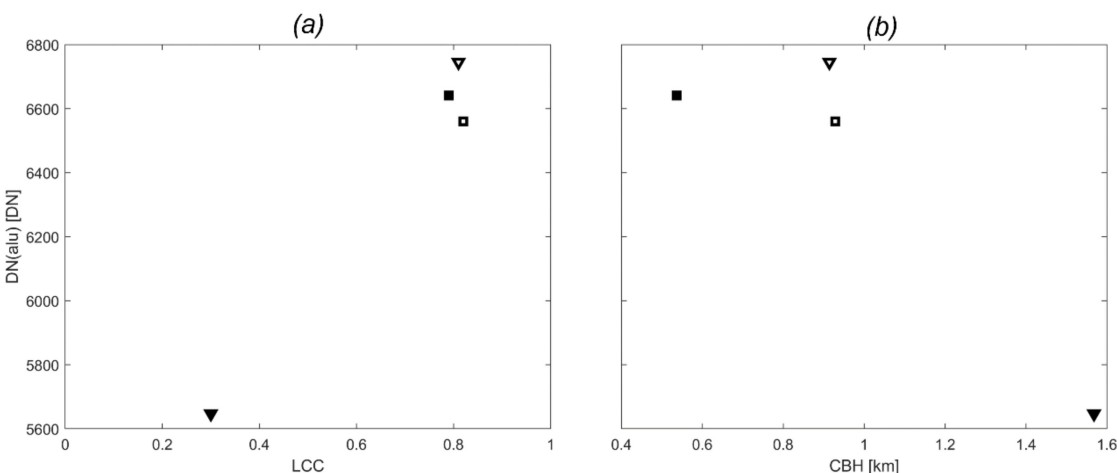

**Figure 6.** Scatter plots of DN(alu) from images after flatfield correction with sigma = 30, as a function of (**a**) LCC, and (**b**) CBH; black squares/triangles indicate April/August field surveys with open/filled symbols representing day/night.

*3.6. Biofouling*

After two weeks, significant biofouling was already visible in VIS and NIR, looking like a brown algal coating, with a mass increase ranging from 0.4% (PET) to 7% (binbag). After three months, biofouling had substantially increased and consisted of, in addition to more brown-green algal coating, brown-green stringy algae, colorless bryozoans, and on the black binbag, small white barnacles. Mass had increased from 8% (PET) to 400% (binbag), compared to virgin items. The biofouling darkened all surfaces in VIS and NIR images, except for white barnacles on the black binbag, which looked bright.

The fact of $T_{air}$ being about 1 °C higher than $T_w$ during the FLIR measurements in the laboratory was recognized in DN(alu) > DN(water). For all wet plastic (biofouled and virgin, except the PET bottle), DN(target) < DN(water). Two weeks of biofouling significantly reduced the DN of wet plastic litter, thereby making the delta value more negative. This could be related to biofouling dampening the TIR reflectance or cooling the plastic surface. We did not find proof that three months of biofouling decreased the delta value of wet plastic further, although we identified more biofouling in the NIR and VIS images. Depending on water and air temperatures, biofouling could enhance or decline the visibility of wet plastic litter in water. For dry plastics, we could not ascertain changes in surface-leaving TIR radiance after two weeks' biofouling but, after three months, there were significant effects. The DN of the black binbag went from zero to >DN(water), indicating increased reflectivity. For dry PET and HDPE bottles, the DN went from >DN(water) to zero, indicating reduced reflectivity. These observations corresponded with the increased and reduced reflectivity in NIR and VIS. We found no relationship between mass increase and changes to DN.

## 4. Discussion

*4.1. Measurements*

The camera response (DN) measured in the laboratory was linear to the surface temperature over limited temperature ranges, with higher intercepts for higher $T_{air}$ (Table 3). Kelly et al. [28] derived a DN = $25T_{bb}$ + 7593 and DN = $23T_{bb}$ + 8748, using a blackbody of temperature $T_{bb}$ for the respective ambient air temperatures of 10 and 21 °C. We believe that a higher intercept for higher $T_{air}$ could be caused by an increased $L_{path}$. The nonlinearity in Equation (7) could be a consequence of increased (lessened) $L_{path}$, due to the presence of warm water (melting the ice) near the FLIR and not the camera's response. Similarly, the slightly higher DN(alu) for $T_{alu}$ (Figure 4), while $T_{air}$ was 19.8 °C, could be explained by the hot brick enhancing the $L_{path}$. According to Kelly et al. [28], during UAV flight conditions, temperature uncertainty in FLIR measurement is ± 5 °C and, during stable laboratory conditions, ± 1 °C. The latter uncertainty was confirmed in our own measurements (Equation (8), RMSE).

Goddijn-Murphy and Williamson [19] followed Peckham et al. [25] in approximating $L_{b,path}$ over the open ocean using $(1 - \tau_{atm})L_b(T_{atm})$, but our results revealed that this was an oversimplification. Goddijn-Murphy and Williamson [19] used $T_{air,2m}$ to estimate $L_{b,sur}$ over the open ocean but, in reality, we do not know how high and wide we need to look or how we can calculate $L_{b,sur}$. The aluminum foil reference target was useful to assess background radiance $L_{b,back}$ ($L_{b,path}$ and $L_{b,sur}$ combined). $L_{b,back}$ appeared to originate in the higher atmosphere, where sub-zero temperatures existed. The presence of clouds also affected the background TIR. In truth, it is difficult to calculate $L_{b,back}$ and, by subtracting $L_{b,w}$ from $L_{b,target}$, we eliminated $L_{b,path}$ (Equation (1)). It follows that by using delta (Equation (6)), the contribution of $L_{b,path}$ was minimized since DN(target) was approximately linear to $L_{b,target}$–$L_{b,path}$ (Figure 3). Delta could be either positive or negative and controlled by water, air, and surface temperatures; the larger its absolute value, the more distinct a target was from the surrounding water. The flatfield correction reduced |delta| somewhat.

Measuring kinetic temperatures correctly has many challenges [30]; our temperature measurement methods could introduce errors that were greater than the given sensor

uncertainties, especially in the field. Logger- or handheld-type thermometers produced significantly different results and we cannot say which sensor was the most accurate. It was, therefore, important to report what type of sensor was used and to keep this in mind when interpreting and comparing the results.

*4.2. Questions Answered*

We evaluate the following questions we set out to answer in Section 1.2, using the results from our experiments, in the following.

**Can a UAV TIR camera operate as a TIR sensor?** The FLIR camera is designed to measure temperature; we recognized this by its response to temperature being near-linear. The relationship between $T_w$ and DN that we derived in the laboratory was quadratic over $T_w$, from $-9$ to 35 °C (Equation (7)) but was linear over smaller temperature ranges (Table 3). It was difficult to estimate the corresponding TIR radiance leaving a surface, due to TIR radiance that has not interacted with the surface ($L_{path}$) and surrounding TIR radiance that is reflected from the surface ($L_{sur}$), captured by the FLIR. It is likely that this background TIR radiance introduced the non-linearity in the camera's response to temperature. In the laboratory, where we could approximate thermal equilibrium, we could estimate $L_{path}$ and $L_{sur}$ by blackbody radiance at room temperature (Equations (2) and (3)). In the field, we recommend imaging an aluminum foil reference target, to assess background radiance. By using delta (Equation (6)), we could reduce the contribution of $L_{path}$. When imaging the ocean from a UAV, we had to account for vignetting (cooler corners) as it was too difficult to keep our targets centered in the view. We could use a flatfield correction to remove the vignetting patterns, but this lessened the DN differences between targets. For estimating delta, we used water and target surface pixels that were close together in uncorrected images.

**How does the TIR radiance transfer model perform?** The TIR radiance transfer model [19] was helpful in interpreting our observations. We learned that using $T_{air,2m}$ to calculate $L_{sur}$ was reasonable in the laboratory but, in the field, we had to look at radiance originating in the higher atmosphere, where there was cold air and the presence of clouds. We need more research to better estimate $L_{path}$ and $L_{sur}$. We also need accurate estimations of the kinetic temperatures of water, air, and plastic surfaces as model input. In the absence of daylight, we can use $T_{air,2m}$ to approximate the surface temperature, $T_{obj}$, of three-dimensional objects and calculate $L_b(T_{obj})$. For two-dimensional items, it is more accurate to use $T_w$. During the day, the absorption of daylight could warm surfaces considerably above $T_{air}$ at the surface, especially dark-colored plastic.

**When and where can we expect the best results?** TIR images of floating plastic and the consequent delta values were different in the absence or presence of sunlight; we can describe two scenarios accordingly:

(A) *Little or no daylight*. At night and in the early morning (surveys 1, 2, and 4) all targets looked cooler than water, reflecting the cold background radiance from the higher atmosphere. The cooler the background radiance, the more negative the DN difference and delta. As the presence of clouds increased the sky's thermal radiance, we saw the largest |delta| under a clear sky. Increased cloud cover and low cloud cover height appeared to reduce |delta| more than warmer air from the surface to a 111-meter altitude. In this scenario, the TIR signal of floating plastic was dominated by the reflectance of cold background radiance, controlled by low cloud cover and cloud base height.

(B) *Daylight*. During survey 3, at around noon, although the sky was overcast, sunlight warmed the targets and all logged kinetic surface temperatures were above water temperature, with some above air temperature. The latter did not include clear plastic bottles, but the targets were possibly not deployed for long enough to see a strong greenhouse effect. The black binbag looked warmest in the TIR image, relating to the enhanced absorption of light by dark colors. In the Aegean Sea survey, the clear PET bottles looked brighter than the binbags [7], this could be because the binbags

were light blue and not a dark colour. White EPS looked the coolest (although the logged temperature was the highest) which would indicate low thermal emissivity. In scenario B, the TIR signal of most floating plastic was dominated by their raised surface temperatures.

In summary, the TIR sensing of marine plastic litter will work best under dark and clear skies (cool-looking plastic), or in the sunshine and warm surface air (hot-looking plastic).

**Which plastic litter items give the best results?** Depending on the conditions, some kinds of plastic litter items were easier to detect in water than others. In scenario A, |delta| increased, from the black binbag, to bottles, to EPS foam board. In scenario B, dark plastic gave the largest |delta|. White EPS foam board stood out as a plastic, with the lowest emissivity (highest reflectivity). This would relate to the relatively high reflectivity (low emissivity) measured for white Styrofoam [20]. The white tarpaulin became too submerged to be visible in TIR in the first three surveys; in the last survey, when the sea was very smooth, the delta value was comparable to that of the blue EPS foam board. According to our laboratory experiments, the delta value was less sensitive to the presence of plastic litter on the water when it was wet. Biofouling on wet plastic reduced the reflectivity and surface temperature. Dense biofouling on dry plastic could enhance or decrease TIR reflectivity, corresponding with the reflectivity in NIR and VIS. Of course, the larger the plastic litter and the more buoyant it is, the more visible it is in TIR under all conditions. We do not expect to directly see microplastics in TIR, as the small particles sink below the water surface and TIR radiance is absorbed in the first 1 mm of water [35].

**Can we separate plastic litter from other surface features on the water surface?** In the TIR images, whitecaps were invisible (Figure A2a), while they were visible in VIS and NIR (Figure A1a,b), which was expected as seafoam emissivity is very close to that of water (Figure 7); presumably, so was its surface temperature. The aluminum foil reference surface looked very different in TIR, due to a very different emissivity; this is promising, in terms of the ability to separate litter like aluminum cans and steel oil drums and needs further research. We added a wooden disk to compare driftwood with man-made plastics, but we did not make enough measurements to establish whether its TIR signal was distinct enough. We did not capture naturally occurring organic material, e.g., floating seaweed in our TIR images; this should be a subject of further study. We did not encounter living sea life, such as surfacing mammals or drifting seabirds, but we expect those to show up as hot spots, especially in scenario A. It may be possible to separate plastic litter from other floating items, and maybe from each other, if we could repeat the aerial surveys in TIR under many more different conditions. For example, the white EPS was the only target (other than aluminum foil) that generated delta < 0 in scenarios A and B. We could relate this to the high TIR reflectivity of white Styrofoam, as measured by Garaba et al. [20]. Their 0.025 μm-resolution hyperspectral TIR spectra of man-made and natural materials that were found along the shore show that the TIR reflectance of the former (e.g., Styrofoam, ropes, and gunny sacks) is higher than the latter (e.g., shells and algae). We could, in theory, separate different materials using spectral features, such as absorption bands, in TIR reflectance. We have measured TIR radiance using a FLIR camera, which operates in the 7.5–13.5 μm spectral band, but findings such as the air, water, and target surface temperature dependence would also apply to narrow bands. When using spectral TIR signatures to identify materials in the field, we need to consider the effects of water, air, and target temperatures on spectral TIR reflectance; we would recommend performing these types of measurements during the night (scenario A).

**Can we see plastic litter on different surfaces than water?** Floating objects stand out in images of the water surface because of three factors:

(1)     Their surface temperature is different from $T_w$.
(2)     Their emissivity is different from $\varepsilon_w$, which is close to one.
(3)     We found $T_w$ to be spatially homogeneous, providing a suitable background.

Regarding (1), water has high thermal inertia, which means that $T_w$ does not change as quickly as does $T_{air}$ and the temperature of plastic. Dry rock and sand have low thermal

inertia and follow $T_{air}$ more closely (Table 4), reducing the temperature and, hence, the difference in plastic litter and land-leaving TIR radiance. However, Lavers et al. [36] found that the moderate (but not low and high) plastic pollution of sandy beaches increased the daily maximum (minimum) by 2.45 (−1.50) °C at 5 cm of depth by altering the thermal inputs and outputs. We could possibly use this finding in the TIR sensing of plastic pollution levels on beaches. Cagnazzo et al. [37] recognized beach litter in thermal images as hotter spots in conditions of air that was warmer than the wet, sandy soil.

Regarding (2), ground with an equally high emissivity (low reflectance) as water should provide a good background for the TIR-sensing of plastic litter. Ice has a thermal emissivity that is similar to or lower than liquid water, with $\varepsilon$ of 0.97–0.98, while for dry sand, $\varepsilon$ is 0.93 for 8–14 μm [24] and is, therefore, less promising, specifically in scenario A. The TIR reflectance spectra of a wide range of surfaces, such as rocks, soil, vegetation, and water, are available from the Jet Propulsion Laboratory's ECOSTRESS spectral library [38–40]. We calculated $\varepsilon$ as $1-\rho$ (assuming opaque materials), averaged over 7.5–13.5 μm (Figure 7). These show that the emissivity of snow is higher than $\varepsilon_w$ and of grass, which is similar to $\varepsilon_w$, making these backgrounds suitable for the TIR sensing of plastic litter. Dry sand has lower emissivity and is therefore a less promising background.

Regarding (3), we do not know how the surface temperature patterns of snow, ice, sand, and other land surfaces can be distinguished. We observed in our FLIR images of the beach that sand can look patchy in TIR images, due to the puddles of water.

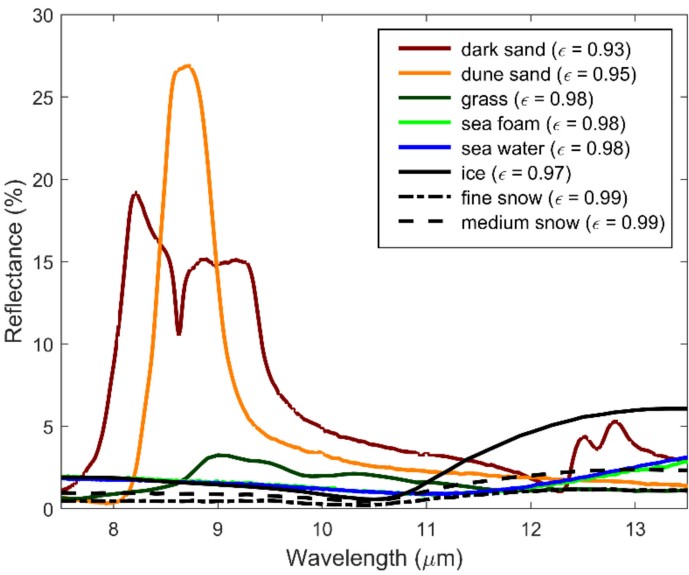

**Figure 7.** Spectral reflectance from 7.5–13.5 μm of brown to dark brown sand, white gypsum dune sand, green rye grass, seafoam, seawater, ice (water), medium granular snow, and fine granular snow from the ECOSTRESS spectral library [38–40]. Mean $\varepsilon$ is shown in the box.

**How can TIR sensing complement VIS-NIR-SWIR sensing?** A valuable special feature of TIR sensing was that no external source of radiance, such as the sun, was needed. The night surveys and the early morning low-light-level survey gave excellent results. We observed additional useful qualities of TIR sensing. As expected, the sea bottom was clearly visible in VIS but not in the NIR and TIR images. This implied that NIR and TIR sensing could help to detect floating plastic litter in shallow coastal waters, where VIS sensing can struggle. Seafoam was clearly visible in our VIS and NIR images, but not in TIR. This was supported by the spectral reflectance of sea foam being close to that of water (Figure 7) and implied that TIR sensing could help separate plastic from oceanic whitecaps, both seen in VIS-NIR, provided that the spatial resolution is high enough. Clear and dark-colored plastics, which can be a challenge in VIS-SWIR sensing, were seen in TIR images.

## 5. Conclusions

We have shown the previously unconfirmed potential of using TIR sensing for floating plastic litter. A UAV TIR camera could monitor floating plastic litter at sea by imaging surface-leaving TIR radiance. Our findings will also, of course, apply to TIR remote sensing from fixed-ground stations (e.g., from a bridge over a river) and they can be used as a starting point for investigating the detection of plastic litter from plane- and space-based TIR sensors. However, the resolution of the latter is currently lower than that of VIS-SWIR satellite observations, and the intervening atmosphere needs to be taken into account. Different scenarios (identified by water, air, and plastic surface temperatures, light intensity, and the presence of clouds) produced different relationships between the radiometric response of the camera and the plastic litter surface in view. This complicated the relationships but could also bring unique opportunities, such as the ability to use the contrast between day and night measurements, although this would be limited in the sea due to the dynamic condition of the floating accumulations. More surveys under a range of different environmental conditions are needed to fully explore this. TIR sensing could complement VIS-SWIR sensing in several valuable ways. For example, TIR sensing could be used during the night, and to detect plastics invisible to VIS-SWIR.

**Author Contributions:** Conceptualization, methodology, formal analysis, data acquisition, writing—original draft preparation, review and editing, and funding acquisition, L.G.-M.; methodology, data acquisition, writing—review and editing, and funding acquisition, B.J.W.; data acquisition, J.M.; review and editing, P.C. All authors have read and agreed to the published version of the manuscript.

**Funding:** This research was funded by the Discovery Element of the European Space Agency's Basic Activities, grant number 4000132579/20/NL/GLC, and aspects of this research were also funded by a Royal Society Research, grant number RSG\R1\180430.

**Data Availability Statement:** Data of the field surveys has been made available through Ocean Scan, https://www.oceanscan.org (accessed on 13 April 2022).

**Acknowledgments:** The authors would like to acknowledge the administrative support of Barbara Bremner and the work of field assistants James Slingsby, Henk Pieter Sterk, Patrick Murphy, Marilou Jourdain de Thieulloy, and Nicholas Petzinna.

**Conflicts of Interest:** The authors declare no conflict of interest.

## Appendix A

*Appendix A.1. Snapshots*

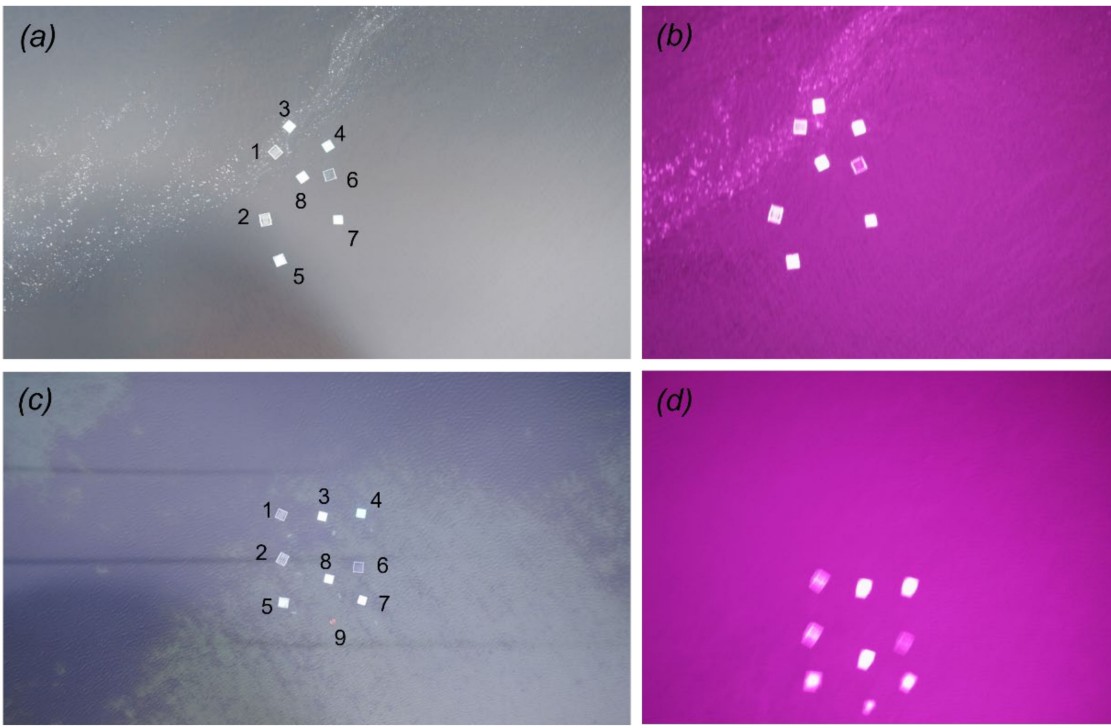

**Figure A1.** Images taken during the four surveys in (**a**) 1 April, RGB, (**b**) 1 April, NIR, (**c**) 3 August, RGB, (**d**) 3 August, NIR. Numbers indicate targets: 1-small PET, 2-large PET, 3-EPS white, 4-EPS blue, 5-HDPE, 6-binbag, 7-tarpaulin, 8-aluminum, 9-wooden disk.

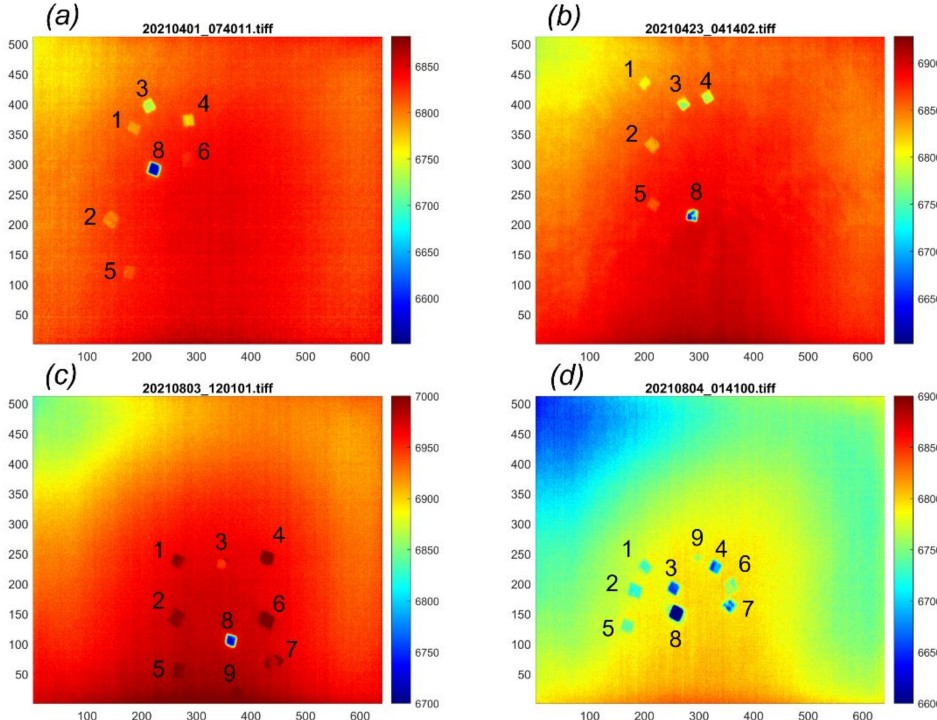

**Figure A2.** Images taken during the four surveys with the FLIR camera, (**a**) 1 April, (**b**) 23 April (**c**) 3 August, (**d**) 4 August; the pseudo color scale indicates DN. Numbers as in Figure A1.

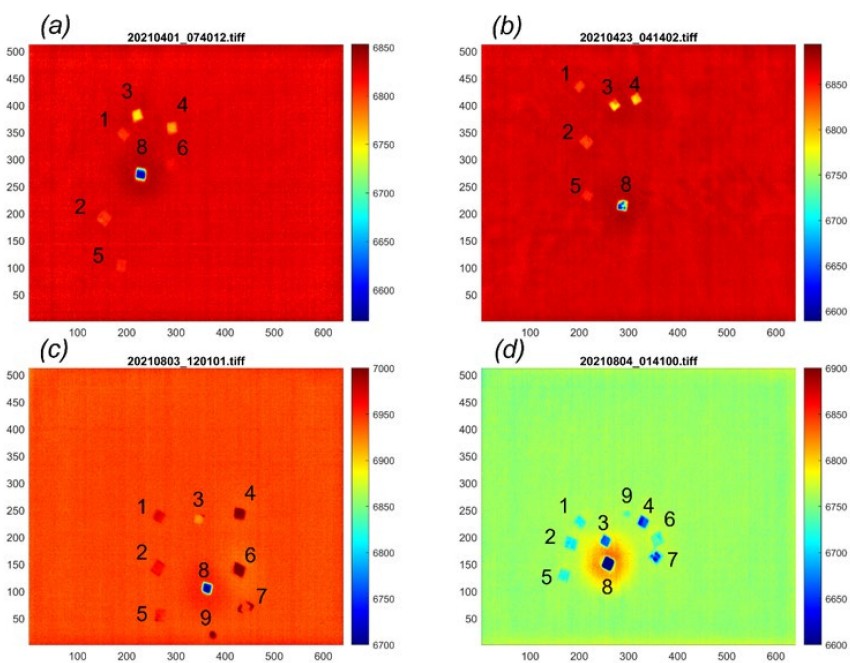

**Figure A3.** Images as in Figure A2 but seen after flatfield correction, using sigma = 30.

*Appendix A.2. Temperatures*

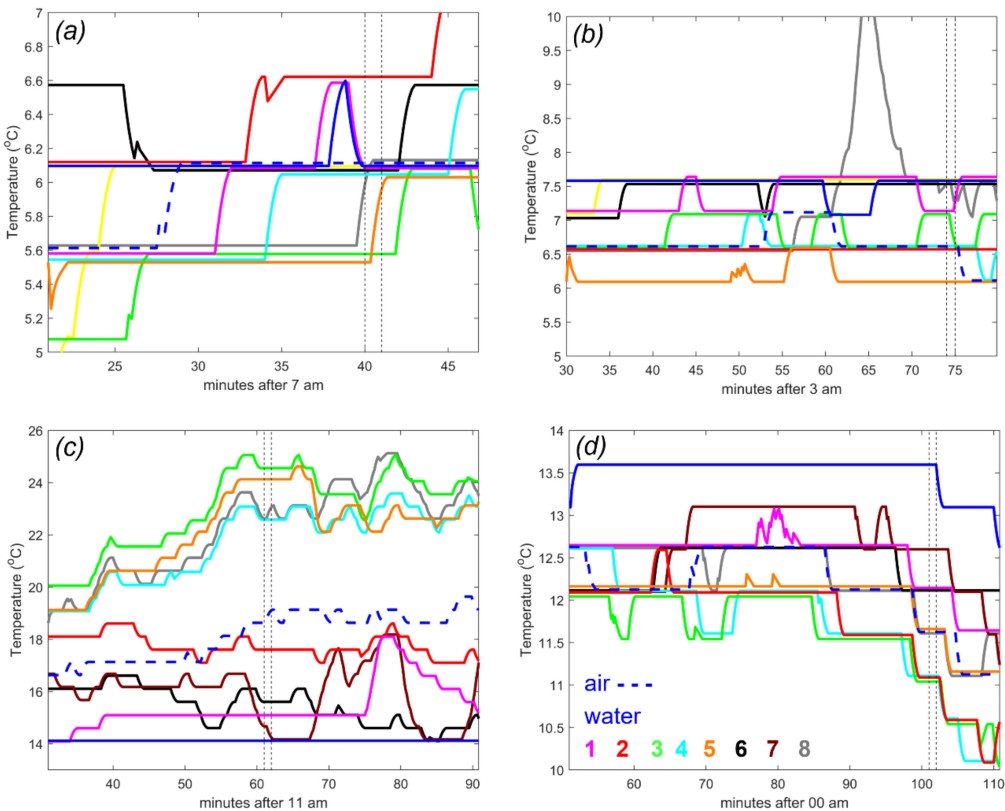

**Figure A4.** Temperature measurements using iButton dataloggers (1-min moving window over 0.1 Hz data) while the targets were deployed at sea for (**a**) survey 1, (**b**) survey 2, (**c**), survey 3, and (**d**) survey 4. Dashed lines indicate the FLIR recording minute and the numbers' targets: 1-small PET, 2-large PET, 3-EPS white, 4-EPS blue, 5-HDPE, 6-binbag, 7-tarpaulin, 8-aluminum.

*Appendix A.3. DN*

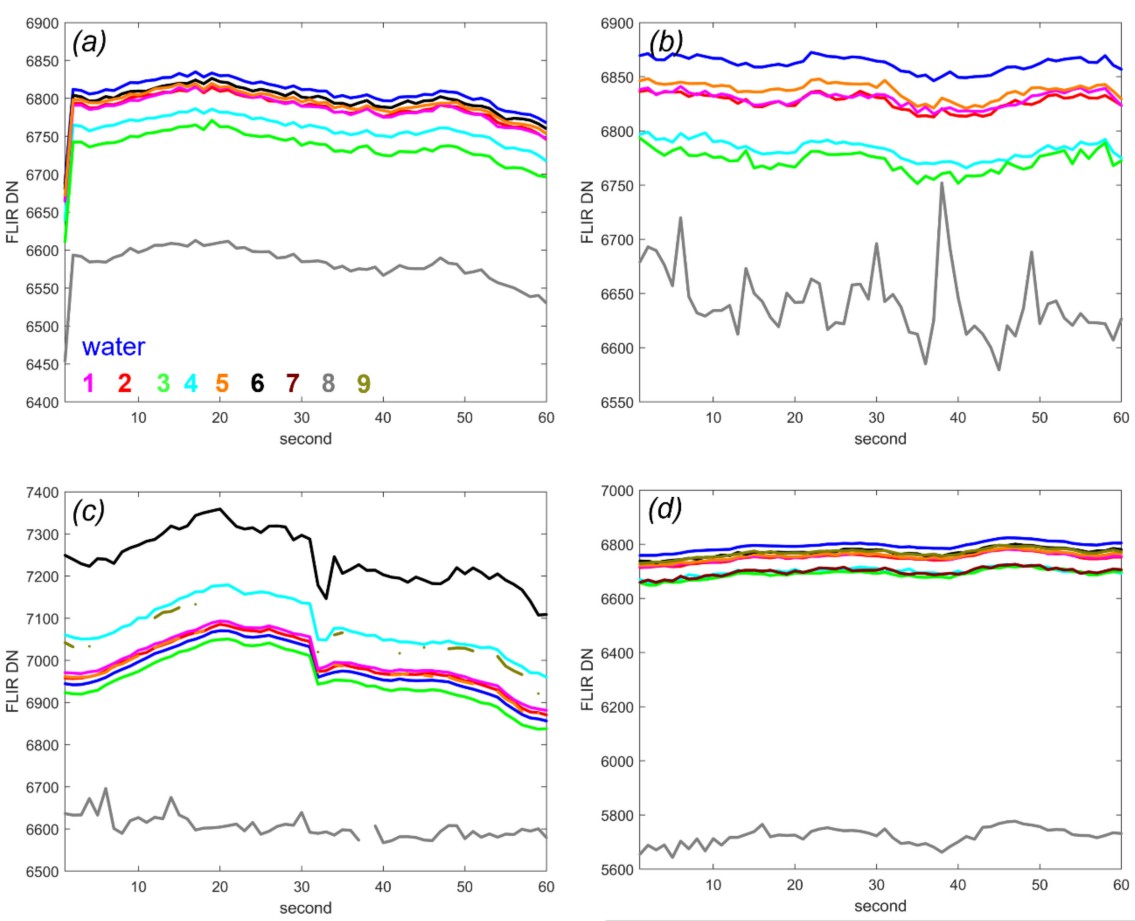

**Figure A5.** DN, 1 Hz data obtained during recording minute, after flatfield correction with sigma = 30, for (**a**) survey 1, (**b**) survey 2, (**c**), survey 3, and (**d**) survey 4. Numbers indicate targets: 1-small PET, 2-large PET, 3-EPS white, 4-EPS blue, 5-HDPE, 6-binbag, 7-tarpaulin, 8-aluminium, 9-wooden disk.

*Appendix A.4. Delta*

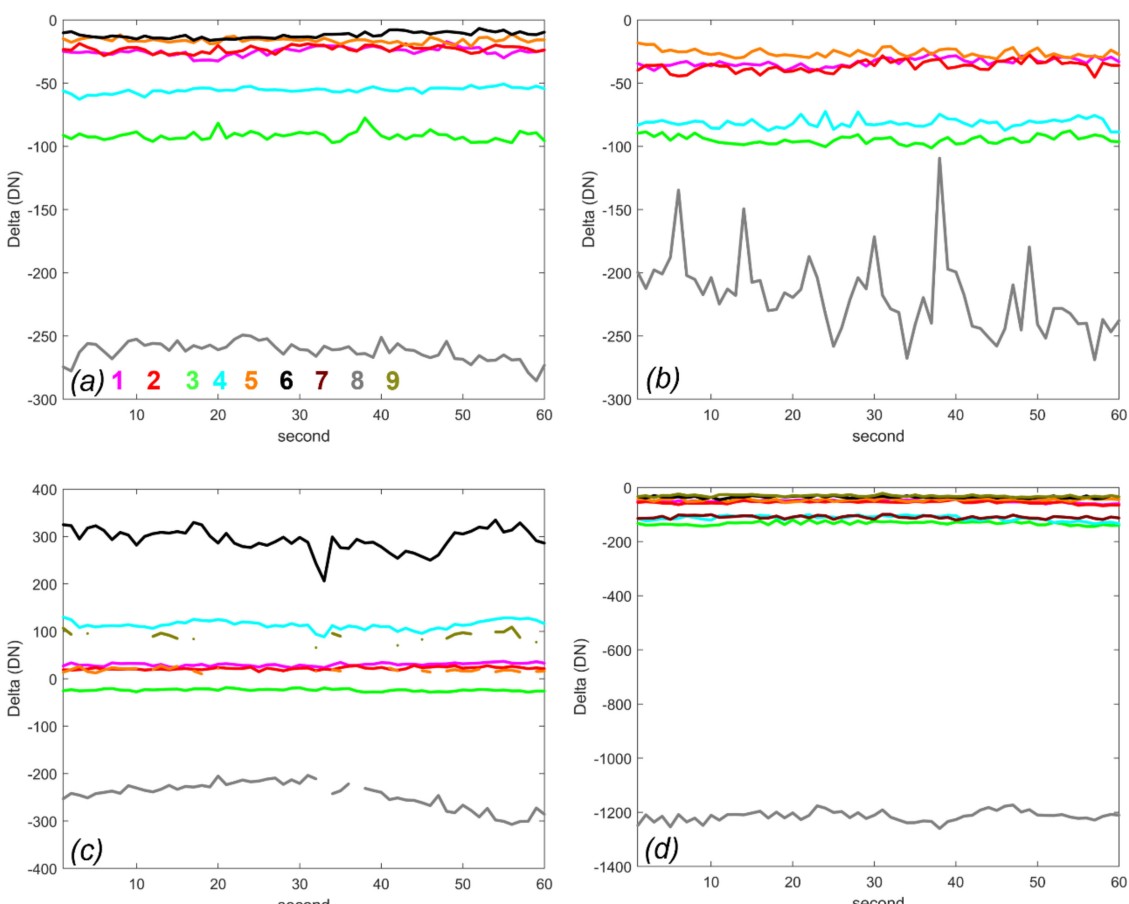

**Figure A6.** Delta, 1 Hz data, obtained during recording minute, using uncorrected images and DN(water) near each target, for (**a**) survey 1, (**b**) survey 2, (**c**), survey 3, and (**d**) survey 4; (derived from Figure A5). Numbers indicate targets: 1-small PET, 2-large PET, 3-EPS white, 4-EPS blue, 5-HDPE, 6-binbag, 7-tarpaulin, 8-aluminum, 9-wooden disk.

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
