# Peer review of "Using a UAV Thermal Infrared Camera for Monitoring Floating Marine Plastic Litter"

_remotesensing, doi:10.3390/rs14133179_

Round 1
Reviewer 1 Report
Dear authors
I've read with a great interest your manuscript dealing on a plastic pollution and the use of UAV thermal infrared camera. I believe such application are very important in particular to obtain more quickly and precise data on marine plastic pollution.
I suggest the authord to add a diagram flow to briefly illustrate the methodology developed and used.
Well done!
Best regards
-------------------------------
I recommend minor revision and in particular for improving the English to make the manuscript more readable.
I believe this is an interesting paper illustrating a new approach to identify marine litter using UAV thermal infrared camera, and especially to monitor the presence of floating litter on the water surface under a range of environmental conditions. Therefore it provides new insights on marine litter monitoring discussing different aspects like the possibility to use UAV TIR camera operate as a TIR sensor, or which plastic items are more identifiable. In addition, the authors also discuss the possibility to use the UAV TIR camera with other surfaces, like on sandy beaches.
However in its current form the description of the method is complex and I suggest the authors to add, for instance, a diagram flow to better illustrate their approach. They also should better illustrate the tests done in situ and the ones in laboratory and to “remove” details that are not directly related to the description of the method in order to make more easily understandable the method. For instance line 221 -222 may be removed. The authors tend to provide too many details on the methods, but maybe some of them can be removed in order to simplify the reading.
I also recommend the authors to read carefully their methods and move eventual results in the result section, like for instance line 181 “The FLIR images showed vignetting, i.e., bright centre and darker on the edges (Fig. A1.2).”; or line “this could produce a brightening artefact seen as ‘auras’ around targets in the images (Fig. A1.3)”.
Similarly I suggest the authors to describe in the result section their results and move discussion element in the discussion section. For instance, l378-379 should be placed in the discussion or l.395-396.
Regarding the discussion, the first paragraph may be removed and I believe the authors should ask a section discussing on the performance and limitation of their approach. It should also be helpful if the authors should discuss the advantage of their method compare to other UAV method for monitoring marine litter. Finally, I recommend the authors to carefully check their manuscript in order to avoid repetition. In addition, some sentences are not clearly linked, which make difficult the reading.
Reviewer 2 Report
A thermal radiance model for floating plastic litter was proposed in this paper.he floating plastic litter targets during the day and night and in different seasons were monitored with a small UAV-grade FLIR Vue Pro R 640 thermal camera. The experiments in the laboratory supported the field measurements in this paper. In general, this paper is very valuable to read.
Review comments:
- As the paper title, “Using a UAV thermal infrared camera for monitoring floating marine plastic litter”, an experimental scheme was proposed mostly. All works and analyzes were based on experimental data. This is a typical practical and experimental paper, which lacks of theoretical originality. For monitoring floating marine plastic litter, the conclusion has nothing to do with the use of UAV. In other words, in the field of remote sensing, there is no direct relationship between the type of observation platform (such as space-based or aviation-based) and the detected Marine plastic waste. If manned aircraft or satellite platforms are used, can TIR sensors with different resolutions also meet the requirements of Marine plastic waste detection?
- Furthermore, in 2.4 Surveys, authors operated the UAV using the DJI GO app to fly UVA in in one position at 30 m altitude. Because of the experiment in open water, how to ensure high reliability detection and stable tracking of Marine plastic waste by UVA flight attitude and route? It is suggested to further explain the experimental environment and UVA platform.
- In Conclusions, the different relationships between the radiometric response of the camera and plastic litter surface were produced by different scenarios. Can build a general qualitative and quantitative model for these relationships? It is important to support the paper Conclusions.
- In fig.6 authors introduced that TIR sensing could help separate plastic from oceanic whitecaps, both seen in OCS. Sea foam was clearly visible in our VIS and NIR images but not in TIR. Seawater and sea foam can be used as clutter background to detect plastic waste in the ocean. However, floating organisms such as water grass at shallow sea will also affect the detection accuracy, which is not reflected in Figure 6. Can give more detail information?
Round 2
Reviewer 2 Report
Author has made definite modification and supplement for the review opinion . It can be accept to published.